# Matrix Information Geometry for Spectral-Based SPD Matrix Signal Detection with Dimensionality Reduction

**DOI:** 10.3390/e22090914

**Published:** 2020-08-20

**Authors:** Sheng Feng, Xiaoqiang Hua, Xiaoqian Zhu

**Affiliations:** 1College of Computer Science, National University of Defense Technology, Changsha 410073, China; fengsh14@lzu.edu.cn; 2College of Meteorology and Oceanography, National University of Defense Technology, Changsha 410073, China; zhu_xiaoqian@nudt.edu.cn

**Keywords:** dimensionality reduction, signal detection, SPD manifold, spectrogram processing

## Abstract

In this paper, a novel signal detector based on matrix information geometric dimensionality reduction (DR) is proposed, which is inspired from spectrogram processing. By short time Fourier transform (STFT), the received data are represented as a 2-D high-precision spectrogram, from which we can well judge whether the signal exists. Previous similar studies extracted insufficient information from these spectrograms, resulting in unsatisfactory detection performance especially for complex signal detection task at low signal-noise-ratio (SNR). To this end, we use a global descriptor to extract abundant features, then exploit the advantages of matrix information geometry technique by constructing the high-dimensional features as symmetric positive definite (SPD) matrices. In this case, our task for signal detection becomes a binary classification problem lying on an SPD manifold. Promoting the discrimination of heterogeneous samples through information geometric DR technique that is dedicated to SPD manifold, our proposed detector achieves satisfactory signal detection performance in low SNR cases using the K distribution simulation and the real-life sea clutter data, which can be widely used in the field of signal detection.

## 1. Introduction

Signal detection is a challenging task in signal processing [1]. As the basic subject in object detection, it is acknowledged as a very valuable research which arouses lots of researchers [2]. However, since complex clutter noise and interference are ubiquitous in the context of signal detection, this crucial task becomes extremely difficult. The main technique in dealing with this issue rely on the Neyman–Pearson criterion [3]. Specifically, by establishing a binary signal statistical detection model, that is, the binary hypothesis $H_{0}$ (no signal exists) or $H_{1}$ (signal exists), the task for signal detection is accomplished with constant false alarm technique, which is referred to as “CFAR”. Although this classical method has made many achievements in some application fields with high efficiency [4], its detection performance still appears a serious bottleneck that limits its practicability, especially for the case of heterogeneous clutter under low signal-noise-ratio (SNR) [5]. Hence, how to achieve satisfactory detection performance under such complex detection background is an issue worthy of attention [6].

From the perspective of time-frequency analysis, known as a powerful technique of research in signal processing [6,7], the sample data processed after short time Fourier transform (STFT) [8] can be exploited for signal detection due to the significant difference between noise and signal. Recent studies have extracted relevant information after STFT which shows that it is feasible for signal detection from the perspective of time-frequency technique [9,10]. More specifically, a signal exists in an area of the spectrogram where the energy change exceeds a certain threshold. In fact, the detection performance of these time-frequency analysis based methods greatly depends on the quality of extracted features from spectrogram. However, former researches only extracted few features maintaining local information, and thus gives inadequate descriptions on the sample data. Meanwhile, their researches are mainly under the prior condition that signal exists, in other words, the problem of false alarm has not been taken into consideration. Hence, such spectrogram processing based methods have many constructive issues to be solved [11].

On the other hand, in the area of matrix information geometry, data representation by symmetric positive definite (SPD) matrix has been widely applied in many scientific researches, e.g., pattern recognition [12], image processing [13], signal processing [14,15] and machine learning [16]. More specifically, by constructing SPD matrices, the original information extracted from the sample data is embedded on a specific SPD manifold, which is shown to outperform the Euclidean space operation. However, the drawback of using these matrix information geometry based methods is the rapidly increasing computational complexity, especially for the high-dimensional SPD manifold [17].

In this case, dimensionality reduction (DR) technique in machine learning is always imperative to reduce the redundancy and improve the discrimination in solving these high computational problems. In fact, simply using conventional Euclidean dimensionality reduction (DR) methods, e.g., Principal Component Analysis (PCA) [18], Linear Discriminant Analysis (LDA) [19], may destroy the implicit structure of these manifold-valued data such that they are unreasonable for our task. Recently, considering the special high-dimensional structure, DR work has been extended to Rienmannian manifold. A generalization of PCA to Riemannian manifold named as PGA [20], which tends to flatten the Riemanian manifold via tangent space mapping, however, does not fully capture the structure of Riemannian manifold, which makes it might be suboptimal for classification. Another popular trend of considering the Reproducing Kernel Hilbert Space (RKHS) mapping by using kernel tricks [21], however, has a huge computational complexity for high-dimensional data, which limits its efficiency. To this end, the information geometric based DR technique dedicated to SPD manifold is proposed [22], by solving a manifold optimization to search a projection matrix, it maps the high-dimension SPD matrices into a lower-dimensional and more discriminative SPD manifold, which has proven its strong power in SPD manifold learning.

The contributions in this paper are following:

According to [23], the high-dimensional feature can provide richer and more discriminative information. In this case, a high-dimensional feature descriptor that contains both local and global information called the dense short time Fourier transform (SIFT) descriptor, is employed to obtain feature vectors of the 2-D spectrograms, which makes better use of information than inadequate edge features previously studied [9,11]. Furthermore, combining with the emerging theory of information geometry, our proposed detector has outperformed the existing spectrogram processing technique based signal detection methods by using information geometric DR method on SPD manifold illustrated in Figure 1. This study presents important information for future development of matrix information geometry-based signal detection methods, which can be widely used for signal detection tasks.

The remainder of this paper is organized as follows: Section 2 introduces some useful backgrounds regarding the SPD Rienmannian manifold and the construction of desired covariance matrix for SPD manifold modeling. Section 3 shows how the novel information geometric DR technique be exploited to our signal detection task. Our experimental results with simulation and real-life sea clutter data are presented in Section 4 and our ongoing works are discussed in Section 5.

## 2. Constructing Region Covariance Matrix on SPD Manifold

In this section, we first provide related knowledge regarding the SPD Rienmannian manifold. Then we show the construction of desired SPD matrix transformed by high-dimensional vector using the 128-dimensional dense SIFT descriptor for feature extraction. So far, the high-dimensional SPD manifold consists of dense SIFT features based SPD matrices is modeled. In other words, the labeled two heterogeneous samples are embed into the constructed SPD manifold.

### 2.1. The Symmetric Positive Definite (SPD) Manifold

In fact, a Riemannian manifold $M_{H}$ with intrinsic geometry structure shown in Figure 2 is a differentiable topological space in which the tangent space $T_{P}$ of each point *p* on the manifold is defined by a smoothly varying inner product. As is well acknowledged, the SPD manifold $S_{++}^{n}$ viewed as the space of n×n SPD matrices, forms the interior of a convex cone in the n(n+1)/2 dimensional Euclidean space. Unlike the linear Euclidean space, most properties and vector operations are not suitable for SPD manifold due to its nonlinearity. In this case, to encode a valid SPD manifold $M_{H}$, appropriate Riemannian metric should be applied so that the similarity between two SPD matrices can be analyzed.

Among these Rienmannian distances designed for SPD manifold, two most popular metrics are Affine-invariant Rienmannian metric (AIRM) [24] and Log-Euclidean metric [25]. More specifically, with the property of affine-invariance, the AIRM metric, which induces the true geodesic distance has been most widely used. However, one of the drawbacks is that the AIRM metric accumulates a high computational burden due to the curvature of such high-dimensional manifold space. Thus, AIRM-based algorithms perform less efficiently than other Riemannian metrics in some engineering cases [26].

On the other hand, induced from Lie group, the Log-Euclidean metric embeds the SPD manifold via matrix logarithm mapping into its tangent space $T_{P}$ so that linear operations can be directly performed. In addition, this Riemannian metric between X and Y enjoys a variety of useful properties [26] that have a simple form of distance measure and low computational complexity, which can be defined as:(1)$$d_{LE}\left(X,Y\right)={∥\log X-\log Y∥}_{F}$$
where log(·) denotes the matrix logarithm operation, and ${∥\cdot ∥}_{F}$ means the Frobenius norm.

### 2.2. The Region Covariance Matrix Based on Dense SIFT Descriptor

The region covariance matrix (RCM) is known as a special kind of SPD matrix with strong robustness and discrimination, which can implicitly capture the second-order statistical characteristics of the sample data. Furthermore, the size of the RCM only depends on the characteristic dimension, hence, regions with different sizes can be directly compared without any transformations. As a consequence of these useful properties, the RCM has achieved remarkable improvement in texture classification [27], pedestrian detection [28], 3D human motion sequence [29], target tracking [30] and many other aspects.

Given a W × H image I with *n* image pixels. The RCM can be given by:(2)$$C_{R}=\frac{1}{n-1}\sum_{k=1}^{n}\left(z_{k}-\mu \right){\left(z_{k}-\mu \right)}^{T}$$
where $z_{k}$ is the characteristic value corresponding to the *k*-th pixel, and $\mu =\frac{1}{n}\sum_{i=1}^{n}z_{k}$ is the average value of the corresponding feature.

In order to fully exploit the information implied in the 2-D spectrograms, the 128-dimensional dense SIFT descriptor with scale invariance is employed for each grid point, which has been shown as a powerful tool in image recognition [31]. The difference between the dense SIFT descriptor and classical SIFT descriptor is that the former is used to extract global image features, while the latter only obtains SIFT features of several key points. Through Gaussian smoothing, the dense SIFT descriptor is obtained by sliding a specific window to record the gradient in 8 directions of each grid point, forming a 4 × 4 × 8-dimensional feature vector shown in Figure 3.

Clearly, using such a global high-dimensional descriptor has its pros and cons: the dense SIFT descriptor can extract abundant information from the sample data but brings unnecessary redundancy, which motivates us to consider effective dimensionality reduction methods.

## 3. Matrix Informantion Geometric Dimensionality Reduction Technique

In order to reduce the redundancy and enhance the discrimination, a manifold optimization based information geometric DR technique is employed. The key idea is to find a projection matrix that maps the original SPD manifold into a low-dimensional one by performing a manifold optimization without changing the intrinsic structure of “the manifold-valued data”. Formally, for a set of SPD matrices $X\in S_{++}^{n}$, our goal is to seek for a mapping $f:S_{++}^{n}\times R^{n\times m}\to S_{++}^{m}$ with the learned projection matrix $W\in R^{n\times m},m<n$. Hence, with the elimination of redundant information by using this technique, “the manifold-valued data” becomes more discriminative, which is shown to overcome the limitations of Euclidean dimensionality reduction methods.

### 3.1. Affinity Graph Embedding

In feature extraction, the training data are generally described by in terms of measurable features that tend to be high-dimensional. In order to reduce the cost of feature measurement, and improve the learning performance, dimensionality reduction that drives new lower-dimensional features from the original features is always employed. However, new feature descriptions invariably cause the discard of the original information, and thus, may give poor overall accuracy. To this end, an affinity function A(i,j) is added in order to force the samples sharing the same label more concentrated, that is, pulling the homogeneous samples closer and pushing the heterogeneous samples away. More specifically, let ${\left\{\left(X_{i},Y_{i}\right)\right\}}_{i=1}^{p}$ denotes *p* labeled spectrograms, here $X_{i}\in S_{++}^{n}$,$Y_{i}\in \left\{0,1\right\}$ represents the label of the two classes. In this case, the affinity function A(i,j) can be established by two nearest neighbor graphs, namely the within-class similarity graph $G_{w}\left(i,j\right)$ and the between-class similarity graph $G_{b}\left(i,j\right)$, which can be defined by: (3)$$\begin{array}{c} G_{w}\left(i,j\right)=\left\{\begin{array}{c} 1,\;\mathrm{if}\;Y_{i}=K_{w}\left(Y_{j}\right)\;\mathrm{or}\;Y_{j}=K_{w}\left(Y_{i}\right)\\ 0,\;\mathrm{otherwise}\; \end{array}\right. \end{array}$$
(4)$$\begin{array}{c} G_{b}\left(i,j\right)=\left\{\begin{array}{c} 1,\;\mathrm{if}\;Y_{i}\neq K_{b}\left(Y_{j}\right)\;\mathrm{or}\;Y_{j}\neq K_{b}\left(Y_{i}\right)\\ 0,\;\mathrm{otherwise}\; \end{array}\right. \end{array}$$

Here, parameter $K_{w}\left(X_{i}\right)$ and $K_{b}\left(X_{i}\right)$ represent the number of the neighbor of $X_{i}$. Hence, the affinity function can be written as: (5)$$\begin{array}{c} A=G_{w}-G_{b} \end{array}$$
which is a binary matrix consisting of 0 and 1. For the balance of $G_{w}\left(i,j\right)$ and $G_{b}\left(i,j\right)$, we usually take $k_{b}<k_{w}$.

### 3.2. Cost Function

In particular, to ensure that the new constructed SPD manifold is valid, that is, $W^{T}XW\;≻\;0$, $\forall X\;\in \;S_{++}^{n}$, an orthogonal constraint $W^{T}W=I_{m}$ is included because the projection matrix $W\in R^{n\times m}$ (m<n) is required to be full rank. After learning the embedding affinity function A(i,j), we confront with the following optimization problem [22]: (6)$$\begin{array}{c} \begin{array}{c} \min\sum_{i,j}A\left(i,j\right)\delta_{lE}^{2}\left(W^{T}X_{i}W,W^{T}X_{j}W\right)\;\\ \;s.t.\;W^{T}W=I_{m} \end{array} \end{array}$$

By substituting Formula (1): (7)$$\begin{array}{c} \begin{array}{c} \min_{W\;\in R^{n\times m}}\;\sum_{i,j=1}^{p}a\left(X_{i},X_{j}\right){∥\log\left(W^{T}X_{i}W\right)-\log\left(W^{T}X_{j}W\right)∥}_{F}^{2}\\ \;\;s.t.\;W^{T}W=I_{m}. \end{array} \end{array}$$

With Talyor expansion that $\log\left(W^{T}XW\right)$ can be approximated as $W^{T}\log\left(X\right)W$, then the optimization problem (7) can be rewritten as: (8)$$\begin{array}{c} \begin{array}{c} \min_{W\;\in R^{n\times m}}\;\sum_{i,j=1}^{p}\;a\left(X_{i},X_{j}\right){∥W^{T}\log\left(X_{i}\right)W\;-\;W^{T}\log\left(X_{j}\right)W∥}_{F}^{2}\\ \;\;s.t.\;W^{T}W=I_{m}. \end{array} \end{array}$$

For simplicity, let
(9)$$\begin{array}{c} \begin{array}{cc} F(W)= & \sum_{i,j=1}^{p}a\left(X_{i},X_{j}\right)\left(\log\left(X_{i}\right)-\log\left(X_{j}\right)\right)WW^{T}\\ \;\; & \times \left(\log\left(X_{i}\right)-\log\left(X_{j}\right)\right) \end{array} \end{array}$$

Then, the optimization problem for finding the projection matrix *W* is given as: (10)$$\begin{array}{c} \begin{array}{cc} \; & \min_{W\in R^{n\times m}}\mathrm{Tr}\left(W^{T}F\left(W\right)W\right)\\ \;\; & \;s.t.\;W^{T}W=I_{m} \end{array} \end{array}$$

In fact, function $\mathrm{Tr}\left(W^{T}F\left(W\right)W\right)$ is independent from the choice of basis spanned by projection matrix *W*, that is, for an arbitrary orthogonal group R∈O(m), $\mathrm{Tr}\left(W^{T}F\left(W\right)W\right)=\mathrm{Tr}\left({\left(WR\right)}^{T}F\left(W\right)\left(WR\right)\right)$, in this case, the solution for Formula (10) known as a Grassmann problem can be optimized by Riemannian Conjugate Gradient (RCG) [32]. However, the RCG method on the Grassmannian manifold converges much more slowly than the iterative two-stage method using eigen-decomposition proposed in [22]. Thus, we take advantage of this eigen-decomposition-based approach for solving Formula (10), which can be summarized as follows. First, F(W) is fixed by assuming that F(W) does not depend on the projection matrix *W*. With the property that matrix trace is equal to the sum of the eigenvalues, the current solution can be obtained by taking the *m* smallest eigenvectors of F(W). Then we have a new *W* to update the corresponding F(W). These steps are repeated until convergence for the optimal solution of *W*, which achieves the mapping from a high-dimensional SPD manifold to a low-dimensional one with more discrimination.

## 4. Experimental Results

To strengthen the algorithm validation, our experiment revolves around simulation and semi-simulation data for signal detection comparing with the state of art techniques. In the simulation data case, we use K distribution to simulate the clutter, and a target signal with Doppler frequency $f_{d}=0.15$ Hz. Additionally, we use the IPIX radar data of sea clutter with 27 cells collected by the Mcmaster University in 1998 [33] for semi-simulation data, likewise, a simulation target with Doppler frequency $f_{d}=0.15$ Hz is added.

In particular, based on the K simulation data, we have in total two labelled spectrogram categories, namely, (a) negative samples in simulation set (no signal exists), (b) positive samples in simulation set (signal exists), which can be clearly seen in Figure 4.

After the short time Fourier transform (STFT), we resized the generated 2-D spectrograms to 200 × 200, using the dense SIFT descriptor to extract feature vector on a regular grid with 2 pixel spacing. We note that the 128-dimensional covariance matrices were constructed by using the method proposed in [34]. In this way, these spectrograms were mapped into an originally 128-dimensional SPD manifold. The overall experimental procedure shown in Figure 5 can be summarized as follows: processed by STFT, the sample data generated a high-precision 2-D spectrogram. Then feature vectors were obtained using SIFT descriptor in feature extraction, which was constructed into SPD matrices for SPD manifold modeling. Finally, a manifold optimization based information geometric DR technique was employed by solving Formula (10) to improve the discrimination, and thus enhance the detection performance.

### 4.1. Simulation Data Experiment

According to our former work in ICSP 2020 [35], we have proved that our proposed approach leads to significant improvement over the original SPD manifold in the case of SNR from −5 to −10, which has shown that the manifold-valued data after information geometric DR become more discriminative, however, we have not explicitly given a practical detector for real-time detection. Furthermore, the signal detection ability of our proposed algorithm in the larger range of SNR is not demonstrated.

To this end, in this paper, we trained one general matrix *W* to form a detector for multiple SNRs by taking all labeled samples into the training set at once rather than training objective *W* for each SNR. More specifically, we had 200 signal sequences per signal-to-noise ratio (SNR −1 to −15) containing the desired signal labelled as the positive samples, and 3000 noise-only samples as the negative samples. To evaluate the detection performance, 4000 signal sequence samples per SNR (−1 to −20) were collected as the test set (2000 positive samples, and 2000 negative samples, respectively), using KNN classifier for efficiency on the low-dimensional and discriminative SPD manifold. Taking false alarm probability $P_{fa}$ and detection probability $P_{d}$ as the two most crucial factors in signal detection as criterion, our experimental results with the K simulation data were the following:

In our first experiment, parameters $K_{w}$ and $K_{b}$ in affinity graph embedding were 40 and 10. The parameter *K* in KNN classifier was fixed to 4200, whose effect on detection performance will be discussed in Section 4.1.2. We first give the detection probability curves of scheme ➀ the Radon Transform detector [36] that were widely used for linear signal detection from spectrogram processing, and scheme ➁ our proposed detector with the objective dimension dim at 40 shown in Figure 6.

It can be seen that the proposed detector outperformed the Radon Transform detector with almost 5 SNR performance improvement. Specifically, our proposed detector led to a significant signal detection capability with a low false alarm rate for the cases with SNR over −10. Since the intra-class matrices constructed under lower SNR cases had little discrimination on the SPD manifold, the detection probability dropped significantly at SNR of interval [−10, −15].

#### 4.1.1. Detection Performance for Various Objective Dimension

In fact, it is obvious that the discrimination of the manifold-valued data was different on various dimensional SPD manifolds, thus the binary classification performance was surely affected. To this end, we compared the detection performance for various objective dimension dim (from 20 to 120 with an interval of 20) on the simulation data, which is shown in Table 1.

The results are shown for SNR of −1, −5, −10, −15, and −20. We note that since the negative samples on the SPD manifold were naturally lying close, there was little difference in false alarm frequency so that the probability $P_{fa}$ for each SNR was almost a constant.

In general, our proposed detector could achieve full signal detection over SNR = −10 in any objective dimension on the simulation dataset. It can be seen that $P_{fa}$ had a proportional correlation with $P_{d}$, and $P_{d}$ decreased rapidly in the SNR range of −10 to −15 because signals at this level became extremely insignificant on the spectrogram. Furthermore, the minimum $P_{fa}$ occurred at dim = 40 while the maximum $P_{d}$ was achieved for dim = 100 but caused a higher false alarm in lower SNR cases.

#### 4.1.2. Parameter Sensitivity Experiment on K

To further control false alarm probability $P_{fa}$, we analyze the sensitivity on parameter *K* in KNN classifier. We selected the case of SNR = −12 with the objective dimension dim = 40 and fixed the parameter $K_{w}$ to 40, $K_{b}$ to 10. Table 2 depicts the detection performance for various values of *K* in the interval [100, 5100].

It can be seen that both the value of $P_{fa}$ and $P_{d}$ decreased with the increase of *K*, which is considered to be because fewer positive samples for each SNR were trained, while the negative samples were relatively large. However, simply increasing the positive samples under each SNR caused more false alarms at low SNR cases because of the small discrimination with the two heterogeneous samples, which also shows that our proposed detector could achieve a satisfactory $P_{d}$ with a very low $P_{fa}$ for SNR over −11 on the simulation dataset.

### 4.2. Semi-Simulation Data Experiment

Simulation data results have shown that our proposed approach was effective, to further evaluate our algorithm, experiments based on semi-simulation data were also conducted. Specifically, in training phase for optimizing the projection matrix *W*, by using IPIX radar data of real-life sea clutter, we had 600 signal sequences per signal-to-noise ratio (SNR 5 to 1) containing the desired signal labelled as the positive samples, and 3000 noise-only samples as the negative samples shown in Figure 7.

Likewise, to evaluate the detection performance, 4000 signal sequence samples per SNR (5 to −10) were collected as the test set (2000 positive samples, and 2000 negative samples, respectively), using KNN classifier on the low-dimensional and discriminative SPD manifold. For the semi-simulation experiment, parameters $K_{w}$ and $K_{b}$ in affinity graph embedding are set to 20 and 4. The parameter *K* in KNN classifier was fixed to 1, known as the nearest neighbor classifier.

In particular, the Radon transform detector misjudged the edge of the clutter as a hidden signal due to the complexity and heterogeneity in our semi-simulation experiment, which showed its drawback in heterogeneous clutter signal detection tasks, thus it was not acceptable for practical application. In this case, we referred to another algorithm as comparison evaluated in our experiments, namely, scheme (a) our proposed information geometric DR based detector, and scheme (b) a mathematical morphology signal detector [37] based on Sobel edge detection.

#### 4.2.1. Detection Performance for Various Objective Dimension

Similarly, as for the real sea clutter data, the impact of the objective dimension dim (from 20 to 120 with an interval of 20) on the detection performance was analyzed shown in Table 3, from which we can clearly see that the detection performance of the proposed detector varied with different dimensions. Generally, the geometric DR detector with lower dimension yielded better detection performance, that is, lower $P_{fa}$ and high $P_{d}$ within a certain SNR range, which indicated that our proposed detector could effectively reduce the redundancy brought by high dimensionality, and thus, improve the detection performance because of its stronger discrimination.

To verify this conclusion, we give the detection probability curves of scheme (a) with the objective dimension dim at 10 and scheme (b) shown in Figure 8. It can be seen that the proposed detector outperformed the Sobel edge detector with Sobel threshold 280. Specifically, scheme (b) gained very limited detection probability $P_{d}$ when considering false alarm $P_{fa}=0.1$. In particular, our proposed detector led to a significant signal detection probability $P_{d}$ with a low false alarm rate $P_{fa}$ for most SNR cases. Since the intra-class SPD matrices under lower SNR cases yielded less discrimination on the manifold, the detection probability $P_{d}$ dropped significantly at SNR of interval [0, −10].

#### 4.2.2. Parameter Sensitivity Experiment on K

In addition, the impact of parameter *K* on detection performance was analyzed by using our proposed algorithm for the semi-simulation data, which can be seen in Table 4. We selected the case of SNR = −5 with the objective dimension dim = 20 and fixed the parameter $K_{w}$ to 20, $K_{b}$ to 4. Table 4 depicts the detection performance for various *K* in the interval [1, 5100]. It can be clearly seen that both the false alarm rate $P_{fa}$ and the detection probability $P_{d}$ increased with *K*, which indicated the same variation trend with the simulation experiment. In particular, to obtain a satisfactory detection performance, a reasonable value of *K* should be taken. For instance, in this part, the false alarm rate $P_{fa}$ could be well controlled by setting a small *K*, that is, the nearest neighbor classifier.

## 5. Discussion

Our work focuses on the task of complex signal detection, establishing a novel signal detection framework based on matrix information geometric dimensionality reduction. The results demonstrate that our proposed detector achieve a satisfactory detection performance in complex signal detection tasks and hence can be widely used in field of signal detection. More specifically, our proposed detector has outperformed the existing signal detection techniques on both the simulation and semi-simulation data with satisfactory detection performance, although the improvement is limited to a certain extent, especially for the real-life sea clutter data. Meanwhile, the constructed SPD matrices in two cases obviously have different distribution on SPD manifold, which indicates that parameters for various signal detection tasks are required to tune for optimum function.

In conclusion, the proposed detector still has some drawbacks required to overcome: The dimensionality reduction algorithm on such high-dimensional signal data constructed SPD manifold has a very high computational complexity, resulting in a large computational cost, which limits the efficiency of our method. In this case, how to accelerate these Riemannian geometry related algorithms is a subject that merits further study. Meanwhile, the form of the constructed SPD matrix has a great influence on detection performance, to this end, appropriate feature descriptors and better SPD matrix construction are areas we intend to study. Additionally, considering the high computing and storage cost on SPD matrices, we choose the KNN algorithm in our experiment, a simple classifier without training process rather than complex predict model-based classification methods, then obtain a satisfactory detection performance. However, the KNN classifier is sensitive to outliers and closely related to the distribution of training samples, which denotes that there might be potential improvement on detection performance. Thus, for future research, we may extend our approach by using other effective classification methods on these spectral-based SPD matrices.

## Figures and Tables

**Figure 1 entropy-22-00914-f001:**
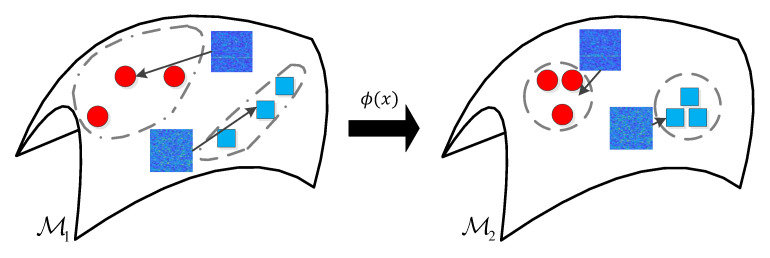
Geometrical interpretation of signal detection method based on information geometric dimensionality reduction. Function ϕ(x) means a non-linear mapping from high-dimensional symmetric positive definite (SPD) manifold $M_{1}$ to lower-dimensional SPD manifold $M_{2}$.

**Figure 2 entropy-22-00914-f002:**
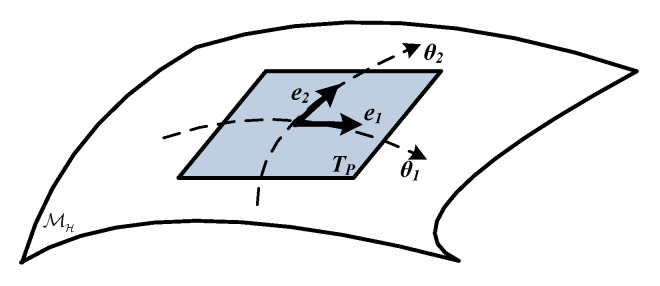
The Riemannian manifold and its tangent space $T_{p}$.

**Figure 3 entropy-22-00914-f003:**
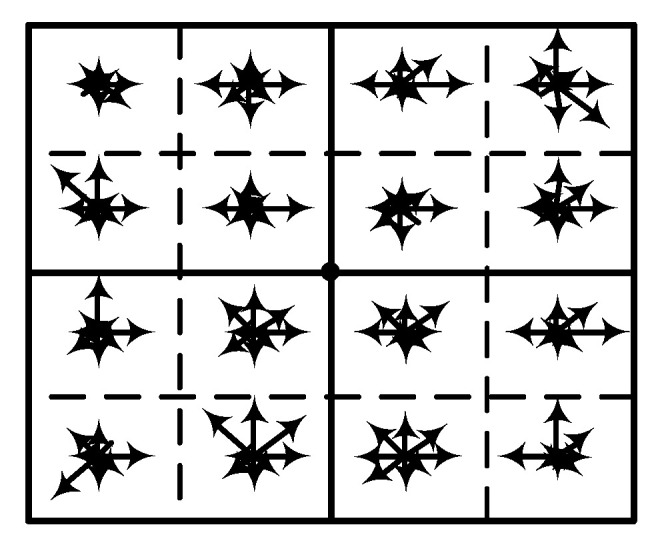
The 128-dimensional dense short time Fourier transform (SIFT) descriptor of one grid point.

**Figure 4 entropy-22-00914-f004:**
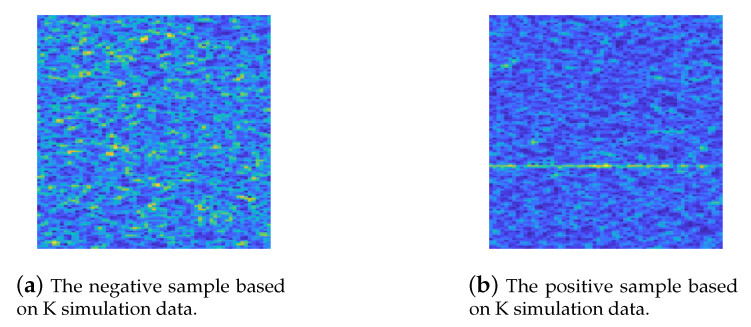
Generated spectrograms on K simulation data with two categaries in case of SNR = −10. (**a**) K simulation data without signal labelled as the negative sample. (**b**) K simulation data containing signal labelled as the positive sample.

**Figure 5 entropy-22-00914-f005:**
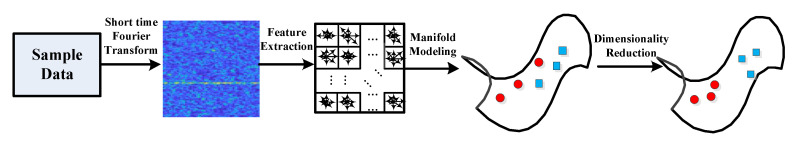
The flowchart of our experiment which can be mainly divided into 4 steps.

**Figure 6 entropy-22-00914-f006:**
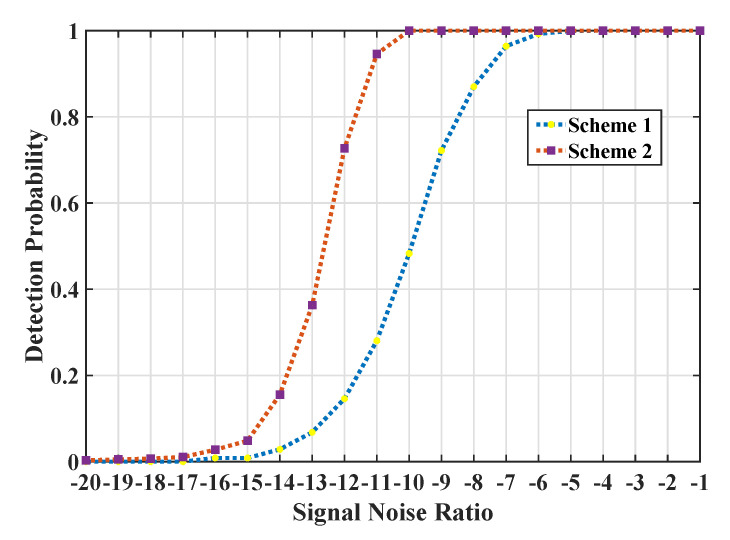
The detection probability curves of the two detectors (scheme ➀ the Radon Transform detector and scheme ➁ our proposed detector at dim = 40 with $P_{fa}$ = 0.001) for varying the signal-noise-ratio based on K simulation.

**Figure 7 entropy-22-00914-f007:**
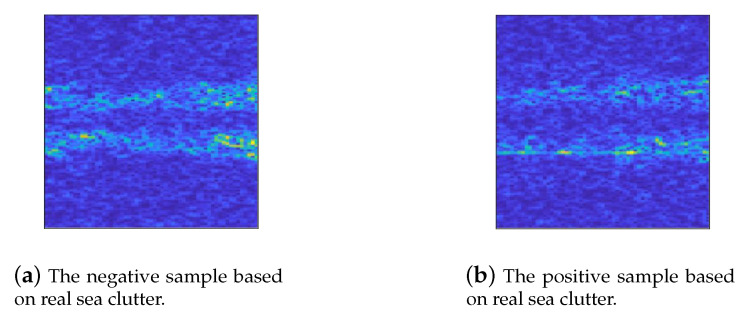
Generated spectrograms on real sea clutter data with two categories in case of SNR = −10. (**a**) Real sea clutter labelled as the negative sample. (**b**) Semi-simulation data containing signal labelled as the positive sample.

**Figure 8 entropy-22-00914-f008:**
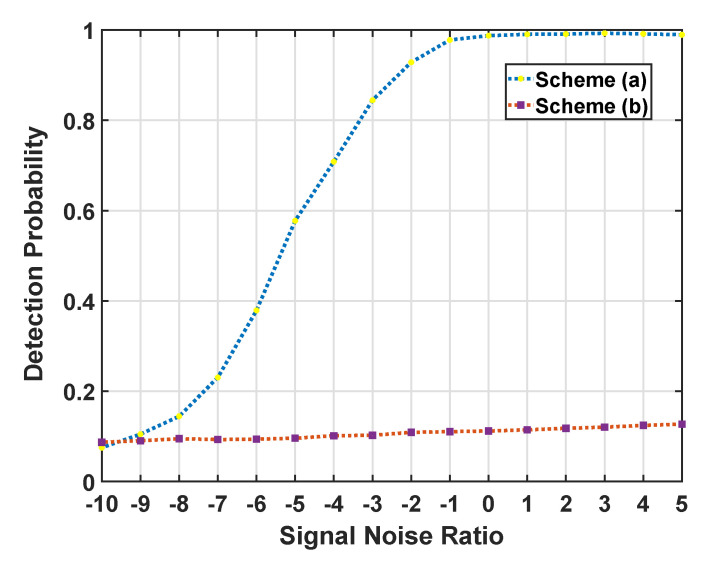
The detection probability curves of the two detectors (scheme (**a**) our proposed detector at dim = 10 with $P_{fa}$ = 5 ×$10^{-4}$ and scheme (**b**) the Sobel edge detector) for varying the signal-noise-ratio based on the real sea clutter.

**Table 1 entropy-22-00914-t001:** Comparison of detection performance for varying the value of dim based on K simulation.

	dim = 20 $P_{\mathrm{fa}}$ = 0.004	dim = 40 $P_{\mathrm{fa}}$ = 0.001	dim = 60 $P_{\mathrm{fa}}$ = 0.0035	dim = 80 $P_{\mathrm{fa}}$ = 0.0085	dim = 100 $P_{\mathrm{fa}}$ = 0.0205	dim = 120 $P_{\mathrm{fa}}$ = 0.02
SNR = −1	1	1	1	1	1	1
SNR = −5	1	1	1	1	1	1
SNR = −10	1	1	1	0.9975	1	1
SNR = −15	0.0925	0.0485	0.1035	0.0085	0.1975	0.1895
SNR = −20	0.0105	0.003	0.011	0.0085	0.0375	0.0355

**Table 2 entropy-22-00914-t002:** Comparison of detection performance at the case of SNR = −12 with the objective dimension dim = 40 for varying the value of *K* based on K simulation.

K	100	1100	2100	3100	4100	5100
$P_{\mathrm{fa}}$	0.016	0.012	0.0085	0.0055	0.0015	0
$P_{d}$	0.9885	0.8875	0.8755	0.8475	0.752	0.4465

**Table 3 entropy-22-00914-t003:** Comparison of detection performance for varying the value of dim based on real sea clutter.

	dim = 20 $P_{\mathrm{fa}}$ = 0.0015	dim = 40 $P_{\mathrm{fa}}$ = 0.002	dim = 60 $P_{\mathrm{fa}}$ = 0.0045	dim = 80 $P_{\mathrm{fa}}$ = 0.0035	dim = 100 $P_{\mathrm{fa}}$ = 0.003	dim = 120 $P_{\mathrm{fa}}$ = 0.0045
SNR = 5	0.989	0.9785	0.9715	0.9645	0.9635	0.9575
SNR = 0	0.9795	0.97	0.9475	0.9415	0.9345	0.921
SNR = −5	0.561	0.575	0.5775	0.526	0.528	0.51
SNR = −10	0.073	0.0765	0.099	0.096	0.102	0.1075

**Table 4 entropy-22-00914-t004:** Comparison of detection performance at the case of SNR = −5 with the objective dimension dim = 20 for varying the value of *K* based on real sea clutter.

K	1	100	1100	2100	3100	4100	5100
$P_{\mathrm{fa}}$	0.015	0.0185	0.086	0.1715	0.253	0.3075	0.4265
$P_{d}$	0.561	0.5985	0.923	0.99	1	1	1

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
