# Peer review of "Matrix Information Geometry for Spectral-Based SPD Matrix Signal Detection with Dimensionality Reduction"

_entropy, 2020, doi:10.3390/e22090914_

Round 1

Reviewer 1 Report

The research is about dimensionality reduction over the Riemannian manifold of SPD matrices and its application in signal detection. The idea of using dimensionality reduction on high dimensional data by removing non-discriminative features could lead to improving system performance. The main comment on this paper is the non-scientific, whiteout reference, and proof description of some scientific concepts.

1- In different parts of the paper, the authors imply somethings without reference or any proof. You can find some examples as follows:

- As a shortcoming of DR techniques approaches the authors implied to the computational complexity of the Kernel-based approach. Which computational complexity do you mean? Please give a reference.

“Another popular trend of considering the Reproducing kernel Hilbert space (RKHS) mapping by using kernel tricks, however, has a huge computational complexity, which limits its efficiency”.

- Which engineer cases. For example, using AIRM is common in EEG signal processing (BCI), maybe most popular. Please mention the cases at least by giving references.

“However, one of the drawbacks is that the AIRM metric accumulates a high computational burden due to the curvature of such high-dimensional manifold space. Thus, AIRM-based algorithms perform less efficiently than other Riemannian metrics in many engineering cases”

2- In different parts, the author used non-scientific keywords. You should use scientific keywords and describe the problem precisely. For example:

Line 61: It is not clear why “high dimensional feature is bound to be powerful”? Also, powerful is not a scientific keyword in machine learning and pattern recognition literature.

3- Figure 1 does not show the Dimensionality reduction on a nonlinear structure. It does not show the change in dimensionality. No description in the text no informative caption. In this figure, you only implied more discriminative representation after DR while it is not the main goal of dimension reduction. You should make it clear both in text and caption of the figure otherwise, this figure is not useful. Some times dimensionality reduction could lead to lower classification accuracy, due to the overlapping resulted from excessive losing the dimensions.

4- How dimensionality reduction could overcome the limitation of Euclidean geometry, and actually what is this limitation? Actually, for analyzing SPD matrices in the Riemannian framework, we have some problems due to the non-linear geometry of the space. Following sentence is a claim without proof.

“Hence, with the elimination of redundant information, “the manifold-valued data” becomes more discriminative, which is shown to overcome the limitations of Euclidean geometry.”

5-You should talk about your objective function, is it convex or not at least you should refer to appropriate reference.

“The above lemma (10) can be solved by an iterative two-stage procedure using eigen-decomposition…”.

6- There are multiple typos and grammatical mistakes in writing. For example:

Line 19: Relys on ,Line33: nanlysis,47: reduncy, Page 3:  applyed, analysed

Reviewer 2 Report

The paper presents a novel approach of signal detection for low SNR cases using a combination of the Short Time Fourier Transform and descriptors. I recommend the publication of the manuscript in the current form.

Author Response

Dear Reviewers:

Thank you very much for your kindly comments on our manuscript (No 880781). Based on your and other reviewers’ suggestions, we have carefully revised the manuscript.

We are now sending the revised article for you. Please see our point to point responses to all your comments below, and the corresponding revisions in the body of the manuscript, both marked in blue. We look forward to hearing from you soon for a decision.

Response to Reviewer 2 Comments:

The paper presents a novel approach of signal detection for low SNR cases using a combination of the Short Time Fourier Transform and descriptors. I recommend the publication of the manuscript in the current form.

We have made further improvements based on your and other reviews' suggestions. Thank you for your time and consideration. And we really appreciate your choice on our article.

Sincerely,

Sheng Feng

Xiaoqiang Hua

Xiaoqian Zhu

Reviewer 3 Report

The paper is devoted to the actual problem of signal detection in noise, the solution of which can be applied in many applications. Many papers have been devoted to this direction and fundamental research has been carried out. Signal detector based on matrix information geometric dimensionality reduction (DR) is proposed. However, the authors of the article need to pay attention to the following questions:

1) The contribution in Introduction is not clearly presented.

2) The quality of signal processing directly depends on the representation of the model of the investigated processes. How do research results depend on the types of noise distribution density, including non-Gaussian noise?

3) A comparison of detectors generally requires ROC curves, and none are given here.

4) The authors used a machine learning approach, but do not explain why the KNN classifier was chosen among all the variety of methods?

5) There are inaccuracies in the text, in particular Table 4: Pfa is 4265 ?

Round 2

Reviewer 1 Report

Hello, 

Thanks for applying comments!

Just few comments. Still exists some typo... please check at least with a English language digital writing tool like Grammarly .

rely on -> relies on

About using the KNN. Classically the main goal of non-linear dimensionality reduction is reducing the dimension with preserving the topology of data as objective function (Isomap, LLE,...). In this space using KNN is a way of evaluation of how much we could preserve the topology in lower dimensional space. So possibly one motivation for using KNN is this fact, it is a classifier which works on local geometry of data and some times we want to evaluate how-much our DR could preserve the local geometry.

Bests

Reviewer 3 Report

The article has been improved and remarks taken into account.

I recommend the publication of the manuscript in the current form.